# Cellular and Molecular Mechanisms of R/S-Roscovitine and CDKs Related Inhibition under Both Focal and Global Cerebral Ischemia: A Focus on Neurovascular Unit and Immune Cells

**DOI:** 10.3390/cells10010104

**Published:** 2021-01-08

**Authors:** Lucas Le Roy, Anne Letondor, Cloé Le Roux, Ahmed Amara, Serge Timsit

**Affiliations:** 1Inserm, Université Brest, EFS, UMR 1078, GGB, F-29200 Brest, France; lucas.leroy3@gmail.com (L.L.R.); anne.letondor@inserm.fr (A.L.); cloe.leroux@hotmail.fr (C.L.R.); ahmedamara90@gmail.com (A.A.); 2Neurology and Stroke Unit Department, CHRU de Brest, Inserm1078, Université de Bretagne Occidentale, F-29200 Brest, France

**Keywords:** ischemic stroke, neurovascular unit, leucocytes, CDK, roscovitine

## Abstract

Ischemic stroke is the second leading cause of death worldwide. Following ischemic stroke, Neurovascular Unit (NVU) inflammation and peripheral leucocytes infiltration are major contributors to the extension of brain lesions. For a long time restricted to neurons, the 10 past years have shown the emergence of an increasing number of studies focusing on the role of Cyclin-Dependent Kinases (CDKs) on the other cells of NVU, as well as on the leucocytes. The most widely used CDKs inhibitor, (*R*)-roscovitine, and its (*S*) isomer both decreased brain lesions in models of global and focal cerebral ischemia. We previously showed that (*S*)-roscovitine acted, at least, by modulating NVU response to ischemia. Interestingly, roscovitine was shown to decrease leucocytes-mediated inflammation in several inflammatory models. Specific inhibition of roscovitine majors target CDK 1, 2, 5, 7, and 9 showed that these CDKs played key roles in inflammatory processes of NVU cells and leucocytes after brain lesions, including ischemic stroke. The data summarized here support the investigation of roscovitine as a potential therapeutic agent for the treatment of ischemic stroke, and provide an overview of CDK 1, 2, 5, 7, and 9 functions in brain cells and leucocytes during cerebral ischemia.

## 1. Introduction

Stroke is the leading cause of disability and the second most frequent cause of death in adults in the world [1,2]. There are two major mechanisms of stroke: intracerebral hemorrhage accounts for about 20% and ischemic stroke for about 80% of cases. Ischemic stroke is mainly caused by obstruction of a cerebral artery, associated with a sudden decrease of regional cerebral blood flow (CBF) and leading to focal ischemia. Two regions according to their perfusion’s level [3] are identified: the ischemic core (CBF ≤ 6 cm^3^ 100 g^−1^ min^−1^) and the penumbra (7–20 cm^3^ 100 g^−1^ min^−1^). In the ischemic core, the cells die quickly, mostly by necrosis [4]. Around the ischemic core, brain cell death progresses more slowly in a heterogeneous, underperfused area called the penumbra [5]. In this region, cells die predominantly by apoptosis [4,6,7]. Both regions are associated with local inflammation. Rescuing the penumbra by recanalization, using either tissue plasminogen activator (tPA) or thrombectomy or both, is the only accepted approach at the acute phase of ischemic stroke. Due to the narrow therapeutic window and the need for imaging before treatment, only 10% to 15% of patients [8,9] are eligible for such treatments.

In addition to cellular death, life-threatening brain edema occurs in about 5% [10,11,12,13] of patients with acute supratentorial infarct [10,11,12] with a case fatality rate of nearly 80% [14,15]. Edema has three components: cytotoxic, ionic, and vasogenic [16,17]. Cytotoxic edema takes place in the ischemic core and is due to energy failure, leading to Na^+^/K^+^-ATPases pump dysfunction followed by osmotic imbalance responsible for cell swelling [17,18,19]. Cytotoxic edema is not accompanied by brain swelling, since water just moves from the extracellular compartment to the intracellular compartment [20]. However, cytotoxic edema generates a force that causes an increase in brain volume: ionic edema [17]. The influx of ions and water into the cells during cytotoxic edema depletes ions from the extracellular space, generating a new osmolar gradient between the extracellular space and the blood circulation [21,22,23]. Therefore, ions and water pass through the blood–brain barrier (BBB) without damage causing an accumulation of extracellular fluid and an increase in brain volume [16,19]. Vasogenic edema, due to BBB disruption, induces water extravasation and is responsible for brain swelling and intracranial hypertension. No medical treatment has proven effective, and decompressive surgery, i.e., hemicraniectomy, is the only option [24,25]. Until now, clinical trials on neuroprotection have not shown efficacy of pharmacological treatment after ischemic brain injury. This failure may be due to a cell-specific targeting effect (for example, targeting only neurons), or a too-specific pharmacological targeting transduction pathway in a disease caused by very different mechanisms involving different cell types [26].

The Neurovascular Unit (NVU) is the subject, in recent decades, of an increasing number of studies highlighting its key role in ischemic stroke [27]. The NVU is described as a “system” responsible for the control and modulation of local cerebral blood flow to adjust optimally local blood flow to neuronal needs [28]. Its function was further extended to different other roles including BBB permeability, immune watch, leucocytes infiltration control, and glymphatic system [29,30,31,32,33,34,35]. The NVU is composed of neurons, astrocytes, microglia, endothelial cells, myocytes, pericytes, and extracellular matrix [36,37]. In ischemic stroke, components of the NVU react, according to the severity of ischemia, to ischemia by a coordinated response [29]. At the acute phase of ischemic stroke, the NVU components—i.e., microglia, astrocytes, or endothelial cells—plays mostly a deleterious role, favoring inflammation and apoptosis (see Section 3). In addition, ischemic stroke increases BBB permeability and endothelial cell adhesion molecule expression, favoring leucocytes infiltration. Leucocytes further enhance the inflammation of brain parenchyma, increase both cell death, BBB disruption, and vasogenic edema (see Section 4).

Since immune cells such as macrophages, neutrophils, and lymphocytes T and B are determining factors for the extension of cerebral ischemic lesions, they should be considered in this review on roscovitine.

Targeting NVU inflammatory-associated processes or leucocyte-mediated inflammation following ischemic stroke was recently studied and represents a therapeutic opportunity. Among potentially promising drugs, cyclin-dependent-kinase inhibitors are particularly relevant because of their pleiotropic action on both NVU and immune cells [38,39,40].

Cyclin-dependent kinases (CDKs) are a family of protein serine/threonine kinases that plays a key role in cell cycle progression in association with non-catalytic regulatory subunit called cyclins. It has 20 members (CDK1-20) described in humans (and other animals) and 13 cyclins (A, B, C, D, E, F, G, H, J, K, L, T, and Y) [41,42]. These enzymes (CDKs 1–6, 11, 14–18) drive each of the major cell cycle transition points (G1, S, G2, M) by phosphorylating selected proteins. CDK protein levels remain stable throughout the cell cycle, but the levels of their regulators (cyclins and specific kinase inhibitors) cyclically vary and, thus, periodically activate CDKs [42,43]. Progression through each cell cycle phase requires different CDK/cyclin pairs. The variations of cyclins gene expression and their destruction induce CDK activity oscillations, which drives the cell cycle [42]. CDKs are also involved in many other cellular functions, including but not limited to transcription (CDK7-13/19/20), apoptosis (CDK1/2/5) differentiation (CDK2/5), DNA repair (CDK1/3/9/12), epigenetic regulation (CDK1/2/4), metabolic regulation (CDK5/8), spermatogenesis (CDK16), and neuronal functions (CDK5/16) [38,42,44]. Among the CDKs family, CDK5 was first identified for its homology with other CDKs but represents an atypical CDK member [45,46,47,48]. In contrast to other CDKs, CDK5 does not require cyclins for its activation but is activated upon association with p35 or p39 cofactors. For a long time, it was thought that CDK5 activity was restricted to the nervous system because CDK5 activators p35 and p39 were mainly expressed in post-mitotic neurons [49,50]. CDK5-p35/p39 function in the nervous system has been widely studied. It is implicated in a large number of functions, including neuronal migration, neurite outgrowth, dendritic arborization, axonal elongation, cell cycle control, synapse formation, synaptic plasticity, memory formation, and pain signaling [45,51,52]. However, since the 2000s, there is an increasing number of studies focusing on non-neural tissues’ CDK5 in healthy and pathological conditions, such as epithelial cells, endothelium, testis, pancreatic cells, muscle cells, leucocytes, or tumor cells [51,52,53]. On non-neuronal cells, CDK5 regulates a very large panel of physiological processes, including cell cycle control, myogenesis, apoptosis, angiogenesis, cell adhesion, cell migration, translation, transcription, and vesicular transport [51].

Roscovitine identification, one of the most popular inhibitors of CDKs, was born from the collaboration of Laurent Meijer’s group of the Biological Station in Roscoff and Jaroslav Vesely and Miroslav Strnad’s group at the Institute of Experimental Botany in Olomouc [54,55]. Roscovitine is a member of the family of 2,6,9-trisubstituted purines, as olomoucine and purvalanol. This family acts by direct competition with adenosine triphosphate (ATP) for binding to the catalytic cleft on CDK [54]. Roscovitine exists as two stereoisomers, (*R*)-roscovitine and (*S*)-roscovitine; however, the (*R*)-stereoisomer is the most frequently studied and used (Figure 1a). Roscovitine selectivity was tested on a wide panel of purified kinase. (*R*)-roscovitine half-maximal inhibitory concentration (IC50) values were below 1 μM for CDK1, CDK2, CDK5, CDK7, and CDK9 and IC50 values were in the 1–40 μM range for few other kinases including DYRK1A and extracellular signal-regulated kinases ERK1 and ERK2 (Figure 1b).

For (*S*)-roscovitine, IC50 was below 1 μM for CDK1, CDK2, CDK5, CDK7 and CDK9 and between 1–40 μM range for DYRK1a, ERK1, ERK2 [56,57] (unpublished data for CDK7 and 9). Affinity chromatography was further assessed on immobilized (*R*) and (*S*)-roscovitine to confirm that the drug only binds to a few protein kinases on various rat tissues (brain, heart, lung, testis, spleen, liver, muscle, kidney) [57]. Roscovitine-binding proteins vary from one tissue to another, but both stereoisomers bind the same proteins, except for pyridoxal kinase (PDXK), which is not bound to (*S*)-roscovitine. The same results were observed in adult rat brain following affinity chromatography: both isomers bound CDK5 and ERK2 but only (*R*) isomer bound PDXK [58]. However, (*R*)-roscovitine’s effects on the catalytic activity of PDXK are limited and do not contribute to its anti-proliferative and anti-apoptosis effect [57]. Cytotoxicity assays in vitro on mouse cells also showed that the two isomers have similar effects on survival dose-response curves and cell cycle distribution. Since PDXK is able to trap (*R*)-roscovitine in vivo, it could reduce its effects on other targets.

## 2. Effects on Clinical Neuroscore, Infarct Size, and Edema: Role of Roscovitine and CDKs Specific Inhibition

Both (*R*)- and (*S*)-roscovitine showed a protective effect on global and focal cerebral ischemia models [58,59,60,61] (Figure 2). We will describe the effects of (*R*)- and (*S*)-roscovitine on neurological score, infarct size, and cerebral edema. In order to understand their global effects, we will study the effect of specific inhibition of roscovitine’s major targets, CDK1, -2, -5, -7, and -9.

### 2.1. Neurological Recovery

#### 2.1.1. Roscovitine

##### (*R*)-Roscovitine and Neurological Recovery after Focal and Global Ischemia

(*R*)-roscovitine intraventricular infusion (ICV) 24 h before transient middle cerebral artery occlusion (tMCAo) in male rat model decreased significantly the neurological deficit scores at days 1, 3, and 5 [59]. In a 2 h tMCAo rat model, (*R*)-roscovitine, injected intravenously (IV) 15 min before MCAo, decreased neurological score (mNSS) 22 h after reperfusion compared to vehicle [62]. (*R*)-roscovitine showed a neuroprotective effect on a 1h mice tMCAo model [61]. Intraperitoneal injection (IP) of (*R*)-roscovitine 15 min before surgery and 1- and 3-h post-surgery decreased neurological score 24 h post-MCAo compared to the vehicle-treated group. In a rat model of chronic cerebral hypoperfusion established by permanent bilateral common carotid arteries occlusion, ICV (*R*)-roscovitine administration 1 day before insult significantly improved behavioral deficit in Morris water maze test 7 and 14 days after surgery [63]. To our knowledge, no studies post-ischemia were performed with R-roscovitine.

##### (*S*)-Roscovitine and Neurological Recovery after Focal Ischemia

Rousselet et al. [60] studied (*S*)-roscovitine effect in a randomized blind study on a 90 min tMCAo rat model. (*S*)-roscovitine administration 15 min post-reperfusion by intravenous (IV) bolus followed by 48 h subcutaneous (SC) infusion improved neurological score 48 h after reperfusion. In a 90 min tMCAo rat model, Le Roy et al. [40] also observed an improved recovery 48 h after reperfusion in (*S*)-roscovitine group compared to vehicle.

#### 2.1.2. Specific CDKs INHIBITION on Neurological Recovery

Specific inhibition of either CDK1 or CDK5 showed a beneficial effect on neurological score following ischemic stroke. In a tMCAo mice model, Marlier et al. [61] observed that CDK1-cKO mice had a lower neurological score compared to WT. Interestingly, neurological score was significantly lower in (*R*)-roscovitine-treated mice compared to CDK1-cKO. Those results suggested that the beneficial effect of (*R*)-roscovitine was in part not due to CDK1 inhibition. In a tMCAo rat model, CDK5miR intra-hippocampal injections during ischemia increased motor recovery from day 1 to 7 after ischemia [64].

#### 2.1.3. Conclusions

(*R*)-roscovitine improves neurological recovery after injection pre-ischemia, and (*S*)-roscovitine improves neurological recovery after injection post-ischemia. Both CDK1 and CDK5-specific inhibition improves neurological recovery.

### 2.2. Infarct Size

#### 2.2.1. Roscovitine

##### (*R*)-Roscovitine and Infarct Size in Focal Ischemia

Beneficial effect of roscovitine on infarction was conducted with (*R*)-roscovitine. In a 1h tMCAO female rat model, (*R*)-roscovitine ICV injection 2 h before MCAo reduced mitosis-specific marker MPM-2 in frontoparietal cortex and reduced phosphorylated tau (PHF-1)-positive cell number in the ischemic cortical regions [65]. (*R*)-roscovitine ICV infusion 24 h before tMCAo in male rat decreased significantly the infarct’s volume at day 1 (−55%), day 3 (−53%), and day 7 (−59%) [59]. In a 2 h tMCAo rat model, (*R*)-roscovitine IV injection 15 min before MCAO decreased infarct volume (triphenyl tetrazolium chloride, TTC) by 51%, 22 h after reperfusion compared to vehicle [62]. (*R*)-roscovitine, also decreased the number of degenerated cells, the number of apoptotic cells, and the number of p-tau positive cells in the parietal cortex compared to the vehicle. In a 1 h tMCAo mice model, (*R*)-roscovitine IP injection at 15 min before surgery, 1 h- and 3-h post-surgery, decreased infarct volume 2 h post-MCAo by almost 55% in both rostral and caudal slices, compared to vehicle group [61]. To our knowledge, no study reported the effect of (*R*)-roscovitine post-ischemic treatment on infarct size.

##### (*S*)-Roscovitine and Infarct Size in Focal Ischemia

(*S*)-roscovitine ICV administration to mice 48 h before pMCAo and throughout the duration of pMCAo led to a 28% decrease of infarct volume compared to vehicle-treated animals at 3 h post-occlusion [58]. Systemic administration of (*S*)-roscovitine by two successive IP injections at 15 min prior and 1 h after the occlusion led to a 31% decrease of the total infarct volume at 3 h post-occlusion, showing no loss of neuroprotective effect [58]. For both administration modes, the hypometabolic zone volume, but not the infarct core, decreased in (*S*)-roscovitine-treated animals compared to vehicle. Interestingly, they observed that the increase of CDK5 activity post-pMCAo was prevented by (*S*)-roscovitine treatment, suggesting that the beneficial effect of (*S*)-roscovitine was at least partly due to CDK5 inhibition. (*S*)-roscovitine neuroprotective efficacy was also assessed on two independents blinded studies in a tMCAo rat model. In the first study, a 90 min tMCAo rat model, (*S*)-roscovitine was administered by IV bolus 15 min prior to ischemia followed by three successive SC injections at 15 min prior to and 24 h and 29 h after the occlusion. (*S*)-roscovitine significantly decreased the infarct volume by 30% 48 h after reperfusion [58]. In the second study, a 120 min tMCAo rat model, (*S*)-roscovitine was administered by IV bolus followed by continuous SC infusion performed 135 min after (post-MCA) the occlusion, leading to a significant decrease by 27% of the infarct volume [58]. Rousselet et al. [60] studied (*S*)-roscovitine effect in a randomized blind study on a tMCAo rat model. (*S*)-roscovitine administration 15 min post-reperfusion by IV bolus followed by 48 h SC infusion decreased infarct volume by 21%, 48 h after reperfusion.

#### 2.2.2. Specific CDKs Inhibition on Infarct Size

CDK1 and CDK5 are associated with detrimental effect on infarct size following ischemic stroke.

##### CDK1 and Infarct Size

On a 1 h tMCAo female rat model, Wen et al. [65] observed an increase in CDK1 and cyclin B1 proteins level as well as an increase in its kinase activity 24 h after ischemia. On a 1 h mice tMCAo model, an immunohistochemistry study showed an overexpression of CDK1 and phosphorylated CDK1 (p-CDK1) in the peri-infarct area 24 h after insult, while they were not detected in the healthy area [61]. Using CDK1-cKO mice subjected to tMCAo, the authors observed a beneficial effect on infarct volume. Interestingly, infarct volume was even significantly lower in mice treated by (*R*)-roscovitine at 15 min before surgery, 1 h- and 3-h post-surgery than in CDK1-cKO mice. These results suggested that the beneficial effect of (*R*)-roscovitine on infarct volume was at least, but not only, due to CDK1 inhibition.

##### CDK5 and Infarct Size

CDK5/p25 hyperactivity was observed in ischemic stroke patients, as well as in several mice and rats models of brain ischemia. An increase of CDK5 activity and CDK5 tau phosphorylating activity was observed in a rat postdecapitative global ischemia model [66]. Wen et al. [67] showed in a female tMCAo rat model that transient ischemia induced p35 cleavage in p25 by calpain, leading to CDK5 hyperactivity. CDK5 hyperactivity was also observed in a 3 h permanent MCAo (pMCAo) mouse model, associated with an increase of CDK5/p25 complexes [58]. In a female rat model of embolic middle cerebral artery occlusion (eMCAO), ischemia increased p25 production at both 6 and 48 h after thrombolysis [68]. CDK5 cKO mice subjected to eMCAO showed a decrease of infarct volume compared to wild-type (WT) 24 h after thrombolysis [68]. CDK5 activity was measured on both focal ischemia and global ischemia models in rats, using CDK5 kinase assay on nuclear and cytoplasmic proteins from the hippocampus [69]. In a rat focal ischemia model induced by endothelin-1 injection, CDK5 activity increased at 3, 12, and 24 h after ischemia in both cells’ cytoplasm and nuclei. In a global ischemia four-vessel occlusion (4VO) model, only cytoplasmic CDK5 activity increased over time [69]. In a 2 h tMCAo female rat model, CDK5 and p35/p25 protein levels were increased in the ischemic hemisphere compared to the sham group [62]. In ischemic stroke in humans, CDK5, p35/p25, and p-CDK5 protein expression were upregulated in infarcted tissue [70].

Inhibition of CDK5/p25 hyperactivity showed a neuroprotective effect in several brain ischemia models. In vitro, an oxygen glucose deprivation (OGD) model induced p25 accumulation and cell death in a time-dependent manner [68]. Using slices from CDK5−cKO mice exposed to OGD, a significant cell death decrease compared to WT was observed. In vivo in a tMCAo rat model, CDK5miR intra-hippocampal injections showed a neuroprotective effect [64]. CDK5 downregulation reduced infarct volume in the first week after ischemia. After one and four months, CDK5miR strongly decreased CDK5 and calpain activities upregulation in CA1, accompanied by a decrease of p25 protein level [64,71]. In addition, CDK5 downregulation decreased hippocampal cell degeneration and promoted plasticity by increasing BDNF (brain-derived neurotrophic factor) protein level in the hippocampus at both one and four months. In an endothelin-1 rat model, injection of both global CDK5 Dominant-negative (DNCDK5), cytoplasmic DNCDK5 or nuclei DNCDK5 2 weeks before ischemia decreased infarct volume compared to WT at day 4 [69,72].

#### 2.2.3. Conclusions

(*R*)-roscovitine decreases infarct size after injection pre-ischemia, and (*S*)-roscovitine decreases infarct size after injection pre- and post-ischemia. CDK1 and CDK5 specific inhibition both decrease infarct size.

### 2.3. Edema

#### 2.3.1. Roscovitine and Brain Edema

No anti-edematous effect was reported using (*R*)-roscovitine. In a randomized blind study on a 90 min tMCAo rat model, (*S*)-roscovitine administration 15 min post-reperfusion by IV bolus followed by 48 h SC infusion decreased brain edema by 37% [60]. Le Roy et al. [40] observed on a 90 min tMCAo rat model that (*S*)-roscovitine decreased brain swelling by 50% and decreased BBB permeability.

#### 2.3.2. CDKs Inhibition and Brain Edema

No study, to our knowledge, reported an anti-edematous effect of CDK specific inhibition after brain ischemia.

## 3. Effects of Roscovitine on Neurovascular Unit

Our recent study investigated the cellular and molecular mechanisms implicated in the anti-edematous effect of (*S*)-roscovitine in a tMCAo rat model. With a Neurovascular Unit analysis approach, we showed that (*S*)-roscovitine’s anti-edematous effect was mediated by the protection of endothelial cells and the decrease of microglial proliferation and astrocytes reactivity [40]. Principal Component Analysis (PCA) was used to study the interaction of the different cellular NVU components. PCA showed that (*S*)-roscovitine’s beneficial effect on BBB and brain edema was due, at least in part, to its pleiotropic effect on the NVU. In order to better understand the effect of roscovitine on neurological recovery, infarct size, and edema volume, we will discuss the effect of roscovitine on NVU after brain ischemia (Figure 2). NVU responses to ischemia were examined in individual cell populations and are summarized below. The role of majors roscovitine targets CDK1, -2, -5, -7, and -9 on NVU cells after ischemia is also discussed. Since data on the study of CDKs implication on non-neuronal cells in cerebral ischemia are relatively scarce, we considered data from other experimental models.

### 3.1. Neurons

#### 3.1.1. Pathological Processes

Neurons play a major role in ischemia-induced inflammatory processes. In the ischemic core, neurons die from necrosis and release, into the extracellular space, reactive oxygen species (ROS), danger-associated molecular patterns (DAMPs) such as ATP, high mobility group box 1 (HMGB1), and heat-shock proteins (HSP), which impact non-neuronal NVU cells, spreading detrimental inflammatory processes [73]. In the penumbra, cells die mainly by apoptosis, which has the advantage that the internal content is not released into the extracellular medium and does not generate inflammation [74,75,76]. However, if apoptotic cells are not promptly cleared, apoptosis can lead to secondary necrosis and then trigger inflammation. Neurons are also capable of releasing “help me” signals to get assistance from non-neuronal NVU cells, principally microglia and vascular cells [77]. These “help me” signals comprise cytokines, chemokines, and growth factors, such as CX3CL1, IL-34, FGF2, LCN2, or IgG, and can differentiate microglial activation into a beneficial phenotype, participating in turn in neuronal recovery by neurotrophic factors release [77]. Another major factor that leads to neuronal death in stroke is spreading depolarization, which comprises repetitive sequences of depolarization/repolarization, from the ischemic core to the penumbra [78]

#### 3.1.2. Roscovitine in Ischemic Stroke Models In Vitro and In Vivo

##### (*R*)-Roscovitine and Neurons

In primary cortical neurons cultures exposed to 4 h OGD, (*R*)-roscovitine increased neuron survival and decreased apoptosis [61]. In another experiment, treatment with (*R*)-roscovitine reduced neuronal apoptosis and decreased phospho-Rb expression in an OGD rat cortical neurons model [79]. In primary rat hippocampal neuronal culture exposed to 1h OGD, (*R*)-roscovitine decreased the phosphorylation level of GluN2B on S1284, a subunit-containing NMDA receptor, suggesting that (*R*)-roscovitine could regulate NMDA receptor function in ischemic conditions [80].

In vivo, Wen et al. [67] first reported a neuroprotective effect of (*R*)-roscovitine in a tMCAO female rat model [67]. ICV (*R*)-roscovitine delivery 2 h before tMCAO decreased neuronal tau hyperphosphorylation 24 h after reperfusion. Another team [59] showed on tMCAo male rat model that ICV (*R*)-roscovitine delivery 24 h prior to ischemia decreased significantly the number of TUNEL-positive neurons in the ischemic hemisphere on days 1, 3, and 5. A neuroprotective effect of (*R*)-roscovitine was also observed in hippocampal CA1 pyramidal neurons of gerbils subjected to 5 min global transient cerebral ischemia (TCI) [81]. (*R*)-roscovitine suppressed CDK5 overexpression, Rb phosphorylation, and p-p53 overexpression after TCI [81]. (*R*)-roscovitine also inhibited neuronal CA1 apoptosis following TCI, with a decrease of Bax, PUMA, and active caspase-3 levels [81].

##### (*S*)-Roscovitine and Neurons

In vivo, (*S*)-roscovitine injection before or after 3 h on mouse permanent MCAo model decreased the volume of the hypometabolic zone, measured by TTC staining, compared to vehicle. TUNEL staining and FluoroJade B-labeling showed that this beneficial effect in the penumbra-like region was partly due to neurons degeneration inhibition [58].

#### 3.1.3. Specific CDKs Inhibition on Neurons

(*R*)- and (*S*)-roscovitine effects on neurons can be mediated by the inhibition of different CDKs. There is in vitro and in vivo evidence for involvement of CDKs in neuronal death after ischemic stroke, including CDK1, -2, -5, and -7 but also cyclins and other CDKs binding partners. All the evidence has been summarized [26,61,67,81,82,83,84,85,86].

##### CDK1 and Neurons

In vitro, Marlier et al. [61] studied CDK1 on primary cortical neuron cultures exposed to 4 h OGD. Neuronal cell death was associated with CDK1 expression, while, after OGD, neurons isolated from CDK1-cKO mice showed greater survival and less apoptosis at 24 h than wild type.

##### CDK2 and Neurons

In neocortical neurons culture subjected to OGD, Katchanov et al. [85] observed an upregulation of cyclin D1, an activation of CDK2, and subsequent cytoskeletal disintegration. In vivo, following 4VO in rat, Timsit et al. [87] observed an increased cyclin D1 mRNA and protein expression level in hippocampal neurons committed to die. In resistant neurons, a lower upregulation of cyclin D1 was observed. The neuronal expression of CDK2 and cyclin D1 also increased in brain sections from patients with focal brain infarction [88].

##### CDK5 and Neurons

In vitro, CDK5, p-CDK5, and p35 expression were increased in human neurons exposed to OGD [70]. On rat primary cultures of striatal neurons subjected to 20 min OGD, p25 synthesis was observed in neurons and was prevented by calpain inhibition [68]. In a model of cultured mouse neurons subjected to a 5h hypoxia followed by 1- to 2-h reoxygenation, DN-CDK5-expressing cells showed better survival than GFP-expressing controls [72]. In cerebellar granule neurons (CGNs) cultures exposed to glutamate phosphorylation of cytoplasmic peroxiredoxin 2 (Prx2) by CDK5 inactivated this antioxidant enzyme and lead to neuronal death [69].

In infarcted brain regions of patients, CDK5, p-CDK5, and p35 protein level expression were increased in neurons [70]. Other authors [70] also observed a co-expression of nuclear CDK5 in TUNEL-positive neurons of peri-infarcted region, suggesting CDK5 involvement in nuclear damages. Consistent results were observed in ischemic brain models. In a tMCAo rat model, CDK5 expression increased in neurons at the boundary of infarct after 3 h [89]. In a transient cerebral ischemia model in gerbil, expression of CDK5, p25, retinoblastoma protein (p-Rb), and p-p53 expression increased in nuclei of CA1 pyramidal neurons on days 1 and 2 [81]. In a rat transient forebrain ischemia model, accumulation of p25 and activation of CDK5 were observed in degenerating neurons in CA1 [86]. Injection in rat hippocampus of DN-CDK5 inhibited CDK5 activity, NMDA receptor phosphorylation by CDK5, and cell death in CA1 neurons. Wen et al. [67] showed in a tMCAo female rat model that hyperphosphorylated tau accumulated in cortical neurons of the ischemic area and was associated with aberrant CDK5 activation. Using affinity assays, they observed that CDK5 was strongly associated with tau in the ischemic brain, suggesting a direct role of CDK5 on tau hyperphosphorylation after brain ischemia. In a tMCAo rat model, CDK5miR intra-hippocampal injections attenuated neuronal shrinkage induced by ischemia and inhibited neuronal (NeuN) loss and Bax (bcl-2–associated X) immunoreactivity levels in CA1 one month after ischemia [64]. In a 4VO rat model, Prx2 inactivation by CDK5 phosphorylation led to neuronal death in CA1. Interestingly, similar results were observed in focal ischemia induced by endothelin-1 [69]. In a rat 4VO model, administration of cytoplasmic DN-CDK5 two weeks before ischemia increased surviving CA1 neurons 4 days after insult. No difference was observed using nuclei DNCDK5 [69].

##### CDK7 and Neurons

Expression of cyclin H, but not its partner CDK7, was increased in hippocampal tissue after global ischemia by a rat 4VO model [90]. Cyclin H immunoreactivity was found exclusively in neurons and increased in ischemic neurons compared to controls.

#### 3.1.4. Conclusions

Both (*R*)- and (*S*)-roscovitine have a neuroprotective effect by decreasing neuronal death on several in vitro and in vivo models of ischemic stroke. After brain ischemia, CDK1 in vitro and CDK5 in vitro and in vivo are upregulated and associated with neuronal apoptosis. CDK2 and CDK7 are upregulated in ischemic neurons in vivo; however, their roles are still unknown.

### 3.2. Microglia

#### 3.2.1. Pathological Processes

Resident microglia cells are immediately activated after brain ischemia. Microglia proliferation peak is observed around 48 h post-stroke and can last several weeks [91,92]. In the acute phase of ischemic stroke, microglia activation is predominantly harmful. M1 microglia release pro-inflammatory cytokines such as IL-1β, IL-6, TNFα, matrix metalloprotease (MMPs), ROS, and nitric oxide (NO), leading to neuronal death, endothelial activation, astrogliosis, and increased blood–brain barrier disruption [30,93,94,95]. Microglia inhibition in the acute phase promotes recovery protects brain barrier integrity and reduces cerebral infarct [96,97,98].

#### 3.2.2. Roscovitine in Ischemic Stroke Models

##### (*R*)-Roscovitine and Microglia

In vitro, in BV-2 cell line subjected to OGD, (*R*)-roscovitine inhibited cell cycle progression and production of IL-1b, MIP-1a (macrophage inflammatory protein 1-alpha), and NO [59]. Inhibition of microglia proliferation by (*R*)-roscovitine was also observed in a rat tMCAo model [59]. Interestingly, its effect was associated in microglia with inhibition of cyclins A, B1, and E upregulation and inhibition of IL-1β, MIP-1a, and NO production.

##### (*S*)-Roscovitine and Microglia

In vivo, (*S*)-roscovitine decreased microglia number in a tMCAo rat model 48h after reperfusion [40]. Principal Component Analysis and correlation matrix of NVU component showed that this effect was associated with BBB protection and beneficial outcome after stroke [40].

#### 3.2.3. Roscovitine in Other Non-Ischemic Models

In vitro, in BV2 cells and primary microglia cells incubated with amyloid beta (Aβ), (*R*)-roscovitine treatment decreased lipoprotein lipase (LPL) level and Aβ phagocytosis [99]. In primary rat brain microglia stimulated by lipopolysaccharide (LPS), pre-treatment with (*R*)-roscovitine decreased microglial activation such as proliferation and NO release [100]. Media from LPS-stimulated microglia pre-treated with roscovitine completely inhibited microglial-induced neuronal death.

In vivo, (*R*)-roscovitine decreased activation and proliferation of microglia on other models of neurological injury: traumatic brain injury [100,101,102], axotomized facial nucleus [103], status epilepticus [104], intracerebral dopaminergic grafts [105], and thermal hyperalgesia [106].

#### 3.2.4. Specific CDKs Inhibition in Microglia

Among CDKs inhibited by roscovitine, at least CDK1, CDK2, and CDK5 modulated microglia reactivity.

##### CDK1 and Microglia

In newborn rats, microglia primary cultures exposed to colony-stimulating Factor1 (CSF1), proliferating microglia increased CDK1 mRNA levels [107]. Consistently, ICV injection of CSF1 increased CDK1 and cyclin B mRNA in microglia in mice [107].

##### CDK2 and Microglia

CDK2 was found to regulate microglial proliferation [103]. In macrophage-colony stimulating factor (M-CSF)-stimulated microglia culture, microglia proliferation was accompanied by an increase of cyclin A and cyclin D level 6 h after stimulation and was maintained until 24 h. However, CDK2 and CDK4 levels did not change. Treatment of microglia cells with PA, a specific CDK2 inhibitor, before M-CSF stimulation suppressed microglia proliferation.

##### CDK5 and Microglia

CDK5/p25 deregulation is involved in microglia activation and phagocytosis in vitro [99]. In BV2 cells and primary rat microglia cultures, incubation with Aβ induced activation of microglia and phagocytosis of Aβ. In both cultures, conversion of p35 to p25 and CDK5 activity increased significantly after Aβ incubation. In BV2 cells, transfection of CDK5 small interfering RNA (siRNA) or p35 siRNA before Aβ incubation decreased the level of LPL, a microglia activation marker, and decreased phagocytosis of Aβ compared to controls [99]. Interestingly, p25 overexpression increased LPL expression and Aβ phagocytosis. In vivo CDK5miR intra-hippocampal injections 30 min after rat tMCAo model decreased microglial hyperactivity of about 50% in CA1 region at 1 month after ischemia [64].

#### 3.2.5. Conclusions

(*R*)- and (*S*)-roscovitine inhibits microglia following ischemic stroke as well as in other in vivo neurological experimental models. CDK1, -2, -5, at least, are present in microglia and are involved in several microglia phenotypes including activation, proliferation, and phagocytosis.

### 3.3. Astrocytes

#### 3.3.1. Pathological Processes

Astrocytes respond to ischemia by cell proliferation and astrogliosis [108,109]. At the acute phase of ischemic stroke, reactive astrocytes promote infarct progression [110], exacerbate inflammation via cytokines production [111], compromise BBB function via VEGFa production [112,113], and aggravate cytotoxic edema through stimulation of aquaporin-4 (AQP-4) channels widely expressed in astrocytic endfeet at the endothelial interface [114,115,116].

#### 3.3.2. Roscovitine in Ischemic Stroke Models

##### (*R*)-Roscovitine and Astrocytes

In vitro, in C6 cell line, (*R*)-roscovitine prevented cell death generated by glutamate-induced gliotoxicity [117]. (*R*)-roscovitine-treated cells exhibited a partial reversion of morphological degeneration induced by glutamate. Its glioprotection effect appeared to be dependent on Rac activation, a protein regulating the cytoskeletal activity.

##### (*S*)-Roscovitine and Astrocytes

In vivo, we showed in a tMCAO rat model that (*S*)-roscovitine treatment decreased astrocytes reactivity 48 h after reperfusion. Principal Component Analysis and correlation matrix of NVU component showed that this effect was associated with edema decrease and beneficial outcome after stroke [40].

#### 3.3.3. Roscovitine in Other Non-Ischemic Models

In vitro, (*R*)-roscovitine inhibited astrocyte activation and inhibited cytoskeletal rearrangement required for migration in a scratch-wound injury model [118], a model commonly used to study astrocyte proliferation [119]. He et al. [118] observed that alone, activated astrocytes adopted an elongated shape with long microtubule-containing protrusions toward the wound, while in the presence of (*R*)-roscovitine, they appeared disorganized and displayed a microtubule meshwork that was not orientated. Using DN-CDK5, they observed that the inhibition on protrusion of wounded astrocyte was weaker than with (*R*)-roscovitine. Then, the stronger inhibition of astrocyte activation by (*R*)-roscovitine than DN-CDK5 suggested that (*R*)-roscovitine’s effect may be mediated by inhibition of several CDKs [118]. In rat primary cortical astrocytes culture incubated for 24 h with fetal bovine serum to induce proliferation, (*R*)-roscovitine reduced cell proliferation in a concentration-dependent manner [120].

In vivo, (*R*)-roscovitine prevented astroglial apoptosis and reactive astrogliosis on other models of neurological injury: status epilepticus [121], TBI [120], and chronic constriction injury [122].

#### 3.3.4. Specific CDKs Inhibition in Astrocytes

Among CDKs inhibited by roscovitine, CDK2, CDK5, and CDK9 were associated with astrocytes.

##### CDK2 and Astrocytes

Serum stimulation of rat astrocytes primary culture-induced CDK2 activation but not CDK7 [123]. CDK2 was also involved in astrocytes differentiation from central glia-4 progenitor cells to an astrocytic cell phenotype [124].

##### CDK5 and Astrocytes

On C6 cell line and primary culture of astrocytes, CDK5 RNAi prevented cell death generated by glutamate-induced glucotoxicity, and induced Rac activation and astrocytic stellation [117]. In neuron–astrocyte co-cultures, CDK5 RNAi-astrocytes displayed a stronger inhibition of neuron degeneration than WT astrocytes following glutamate-induced excitotoxicity [117]. This neuroprotection was associated with Rac1 activation and brain-derived neurotrophic factor (BDNF) upregulation in astrocytes. In endothelial cell–astrocyte co-cultures, CDK5-KD astrocytes displayed stronger protection of endothelial cells than WT astrocytes following glutamate-induced excitotoxicity, protecting PECAM-1 and F-actin cytoskeleton and inducing BDNF release in endothelial cells and in astrocytes [125]. In vivo, in rat 2VO model, transplantation of CDK5-knockdown (KD) astrocytes into the somatosensory cortex after ischemia rescued motor and neurological impairment the first week compared to transplantation of WT-astrocytes [126]. Neurological and motor rescue were still significantly better after 4 months [125]. CDK5-KD astrocytes transplantation also stimulated endogenous astrocytes and endothelial cells and increased BDNF protein level 15 days after transplantation compared to WT-astrocytes [126]. At 4 months, CDK5-KD astrocytes transplantation prevented neurons loss, astrocytes loss, and astrocytes hypertrophy compared to WT-transplantation [125]. In addition, CDK5-KD astrocytes transplantation partially prevented BBB disruption 4 months after ischemia. In tMCAO rat model, CDK5miR intra-hippocampal injections during occlusion decreased astrocytes swelling and hypertrophic somas in CA1 one month after ischemia [64].

##### CDK9 and Astrocytes

CDK9 appeared to be involved in astrocytes gene regulation in vitro. CDK9 RNAi transfection in primary human astrocytes was shown to strongly upregulate gene expression [127].

#### 3.3.5. Conclusions

Reactive astrocytes are inhibited by (*S*)-roscovitine after brain ischemia and by (*R*)-roscovitine on other non-ischemic models. CDK2, -5, and -9, at least, are present in astrocytes. Following ischemia, CDK5 is involved in astrocyte reactivity.

### 3.4. Oligodendrocytes

#### 3.4.1. Pathological Processes

Many oligodendrocytes die within 3 h after a stroke [128]. We do not know much about the role of oligodendrocytes at the acute phase of stroke; however, demyelination processes have severe effects on axonal function, metabolism, and survival [128,129,130]. Microglia and astrocyte were shown to enhance demyelination processes by secreting pro-inflammatory molecules and nitric oxide. A recent study showed on a tMCAo mice model that oligodendrocyte progenitor cells (OPCs) transplantation alleviated edema and infarct volume, promoted neurological recovery, and reduced BBB leakage by increasing claudin-5, occludin, and β-catenin expression [131]. Moreover, oligodendrocytes play a major role in post-stroke recovery [132]. Ischemic stroke induces the proliferation and migration of OPCs, which differentiate into mature oligodendrocytes (OL) in order to form myelin, thereby promoting neuronal recovery [130,133,134]. Microglia and astrocyte were shown to modulate oligodendrocytes during remyelination [132]. Both microglia and astrocytes play a dual role in remyelination processes. Pro-inflammatory phenotypes M1 and A1 prevent remyelination, while anti-inflammatory phenotypes M2 and A2 favor it.

#### 3.4.2. Roscovitine

No studies were conducted, to our knowledge, on oligodendrocytes treated by (*S*)- or (*R*)-roscovitine on models of ischemic stroke. However (*R*)-roscovitine effect was tested on other oligodendrocyte injury models. In vitro, on rat cortical OPCs culture, (*R*)-roscovitine strongly inhibited OPCs proliferation induced by platelet-derived growth factor (PDGF) and prevented OPCs apoptosis induced by growth factor deprivation [135]. Other authors showed that (*R*)-roscovitine reduced differentiation and migration of OPCs in vitro [136,137]. Another study showed that (*R*)-roscovitine treatment resulted in the reduction of OPCs maturation but had no effect on OPC cell proliferation [138].

In vivo, in a focal demyelination model induced by lysolecithin (LPC) injection in dorsal spinal cord in mice, myelin repair was significantly impaired in animals that received local injection of (*R*)-roscovitine [139]. CDK5 cKO also showed significantly reduced myelin repair, suggesting that (*R*)-roscovitine’s effect was at least partly mediated by CDK5 inhibition.

#### 3.4.3. Specific CDKs Inhibition in Oligodendrocytes

Among CDKs inhibited by roscovitine, CDK1, CDK2, and CDK5 were studied in oligodendrocytes.

##### CDK1 and Oligodendrocytes

The only study on hypoxic oligodendrocytes showed on rat OPCs maintained at 1% and 4% O2 for up to 7 days that both hypoxia conditions induced an increase in p-CDK1 and p-Rb levels [140].

##### CDK2 and Oligodendrocytes

CDKs functions in oligodendrocytes were more deeply studied in vitro in non-ischemic models. In purified rat OPCs cultured with PDGF, proliferating OPCs had higher protein levels of cyclin E and CDK2 and higher kinase activities of CDK2-cyclin E compared to non-proliferating cells [141]. In differentiated oligodendrocytes, authors observed a decrease in CDK2-cyclin E complexes formation. Consistently, in OPCs purified from adult CDK2^−/−^ and WT mice grown for 5 days in vitro, the number of immature oligodendrocytes was reduced threefold, and the number of pre-oligodendrocytes was increased in CDK2^−/−^ cultures as compared with WT cells [142]. CDK2^−/−^ oligodendrocytes also displayed an increased number of processes. Furthermore, DN-CDK2 transfection on cultured OPCs was shown to inhibit OPC proliferation [143]. CDK2 was also studied in non-ischemic models in vivo. CDK2 was associated with remyelination after focal demyelination in mice [142,144]. In a mice model of focal demyelination induced by LPC injection, CDK2^−/−^ mice showed a significant OPCs proliferation decrease compared to WT mice 7 days after demyelination [142]. Interestingly, CDK2^−/−^ mice showed an enhanced differentiation of immature OPCs to mature oligodendrocytes than WT at 14 days but not at 21 days, suggesting that in the absence of CDK2, the rate of oligodendrocyte differentiation was accelerated during the process of remyelination [142]. At 14 days, the authors [142] observed by electron microscopy a two-fold increase in the percentage of myelinated axons in the lesion of CDK2^−/−^ mice as compared with WT.

##### CDK5 and Oligodendrocytes

CDK5 is expressed in OPCs and oligodendrocytes during multiple distinct stages of development (A2B5, O4, MBP, NG2) [138]. CDK5 RNAi transfection in spinal cord mixed cell cultures showed that CDK5 KD oligodendrocytes presented a reduced process arborizations compared to WT [138]. Conversely, overexpression of CDK5 resulted in highly branched processes, suggesting that CDK5 was involved in oligodendrocyte maturation. In FBD-102b cells and primary rat OPCs, induction of differentiation increased CDK5 protein and mRNA expression, as well as CDK5 kinase activity [136]. CDK5-KD on FBD-102b and primary rat OPCs reduced the differentiation of OPCs into OLs compared to WT. The same author also showed that CDK5 control OPCs’ migration in vitro, and CDK5-KD impaired OPCs migration [137]. In vivo, specific deletion of CDK5 in OLs by CDK5 conditional KO (cKO) significantly delayed myelin repair in LPC-induced focal demyelination in mice dorsal spinal cord [139]. CDK5 cKO also reduced mature CC1^+^ cells but increased Olig2^+^ cells in LPC-induced lesions at 3, 7, and 14 days. The remyelination failure was associated with a reduction of Akt signaling and an enhancement of Gsk-3β signaling pathways.

#### 3.4.4. Conclusions

(*R*)-roscovitine reduces OPCs differentiation, proliferation, and migration in vitro and reduces remyelination after focal demyelination in vivo. CDK1 level is increased following hypoxia; however, its role is unclear. In other oligodendrocyte injury models, CDK2 inhibits OPCs differentiation in vitro, and inhibition of CDK2 in vivo shortens remyelination. Conversely, CDK5 is necessary for OPCs differentiation in vitro, and inhibition of CDK5 in vivo delays remyelination.

### 3.5. Endothelial Cells

#### 3.5.1. Pathological Processes

Endothelial cells (ECs) activation appears to be mainly deleterious at the acute phase of focal ischemia. In a transient cerebral ischemia mouse model, injection of endothelial cell microvesicles, prepared from endothelial cell cultured in oxygen and glucose deprivation conditions, increased infarct volume and neurological deficit score and worsened BBB disruption [145]. Didier et al. [146] showed that endothelial cells could amplify inflammatory cytokines secretion by astrocytes after being activated by TNFα. Activated endothelial cells also expressed adhesion molecules (selectin, ICAM-1 and VCAM-1, integrins) favoring leucocyte infiltration, which exacerbates neuronal damage [147,148,149]. In a model of cerebral ischemia in mice, deletion of brain endothelial IL-1R1 improved cerebral blood flow, reduced neutrophil infiltration, and vascular activation 24 h after brain injury [150].

#### 3.5.2. Roscovitine in Ischemic Stroke Models

##### (*R*)-Roscovitine and Endothelial Cells

In culture of bEnd.3 cells, a mouse brain cell line, exposed to glutamate excitotoxicity [125], (*R*)-roscovitine treatment rescued transendothelial resistance (TEER) disruption and loss of endothelial adhesion protein PECAM-1 and p120. (*R*)-roscovitine also prevented intercellular gaps number increase induced by glutamate.

##### (*S*)-Roscovitine and Endothelial Cells

We previously showed in tMCAo rat model that (*S*)-roscovitine protected ECs permeability, probably by protecting occludin, and also decreased endothelial RECA-1 (rat endothelial cell antigen 1) hyperstaining induced by ischemia [40].

#### 3.5.3. Roscovitine in Other Non-Ischemic Models

(*R*)-roscovitine effect was tested by Berberich et al. [151] on ECs inflammatory model. On human umbilical vein endothelial cell (HUVEC) culture, TNFα was applied; 24 h later, freshly isolated human granulocytes were added for 30 min. Then, HUVECs were washed to remove non-adherent granulocytes. (*R*)-roscovitine pre-treatment decreased granulocytes adhesion to endothelial cells in a concentration-dependent manner. (*R*)-roscovitine also inhibited, in a concentration-dependent manner, the increase of surface adhesion proteins ICAM-1, E-selectin, and VCAM-1, as well as the ICAM-1 mRNA expression. Interestingly, authors observed a CD11b surface-level decrease in human granulocytes. They showed, using kinome arrays and CDK activity panels, that inhibition of CDK5 and CDK9 in ECs was responsible for this anti-inflammatory action. In the HUVECs scratching model, (*R*)-roscovitine decreased ECs migration [152]. Ex vivo in mouse aortic rings, (*R*)-roscovitine inhibited ECs sprouting [152].

#### 3.5.4. Specific CDKs Inhibition in Endothelial Cells

Among CDKs inhibited by roscovitine, CDK1, CDK2, CDK5, CDK7, and CDK9 were studied in endothelial cells.

##### CDK1 and Endothelial Cells

In vitro in HUVECs, CDK1 siRNAs decreased ECs proliferation, migration, and capillary-like tube formation [153]. In vivo in an oxygen-induced retinopathy (OIR) mouse model, CDK1 was overexpressed in ECs [153]

##### CDK2 and Endothelial Cells

In HUVECs culture, TNFα exposure increased expression of surface adhesion molecule ICAM-1 [151]. CDK2 siRNA transfection before TNFα treatment did not prevent upregulation of ICAM-1 by TNFα. In bovine aorta ECs (BAECs) culture, CDK2 kinase activity was upregulated in proliferating cells, compared to confluent cells that were contact-inhibited [154]. Similarly, on a confluent BAECs culture monolayer scraped-injured model to induce cell migration and proliferation, CDK2 kinase activity increased 4 h after injury.

##### CDK5 and Endothelial Cells

Mitsios et al. [70] observed in human brain microvascular endothelial cells culture exposed to OGD an increase of CDK5, p-CDK5, and p35 expression. CDK5 functions in vitro in ECs were further studied in other inflammatory models. In HUVECs culture stimulated with TNFα, CDK5 short hairpin RNA (shRNAs) transfection before TNFα treatment strongly decreased upregulation of ICAM-1, suggesting that CDK5 was involved in ECs activation [151].

In vivo, Liebl et al. [152] showed that CDK5 was expressed by human endothelium. In hypoxic regions of infarcted human tissue, overexpression of CDK5 together with p35/p25 was observed in apoptotic brain ECs and was associated with cellular damage as a response to hypoxic conditions [70].

##### CDK7 and Endothelial Cells

Berberich et al. [151] showed that CDK7 siRNA transfection before TNFα exposure in HUVECs culture did not prevent upregulation of ICAM-1 by TNFα. However in HUVECs culture stimulated with VEGF, other authors [155] observed an enhanced CDK7 expression and an enhanced RNA polymerase II (RNAPII) expression and phosphorylation. THZ1, a selective covalent inhibitor of CDK7, suppressed the proliferation and mobility of VEGF-activated ECs 24 h and 48 h after stimulation. THZ1 also suppressed VEGFR2 expression and RNAPII phosphorylation. Furthermore, THZ1 treatment reduced capillary-like tube formation. Consistently, CDK7 knockdown by siRNA reduced capillary-like tube formation after VEGF stimulation [155].

##### CDK9 and Endothelial Cells

CDK9 shRNA transfection before TNFα stimulation in HUVECs culture strongly decreased the upregulation of ICAM-1 induced by TNFα, suggesting that CDK9 was involved in ECs activation [151]. In another study [156], the same team showed that HUVECs transfection with CDK9 siRNA inhibited both the expressions of ICAM-1, VCAM-1, and E-selectin in response to TNFα.

#### 3.5.5. Conclusions

(*S*)-roscovitine in vivo and (*R*)-roscovitine in vitro protects ECs in ischemic stroke models. In other models, (*R*)-roscovitine inhibits ECs proliferation, activation, migration, and inflammation.

CDK5 expression is increased after ischemic stroke in endothelial cells. In other models, CDK1 and CDK2 are involved in ECs proliferation and apoptosis, CDK5 and CDK7 regulates ECs migration and angiogenesis, and both CDK5 and CDK9 are involved in ECs inflammation.

## 4. Effects of Roscovitine on Leucocytes

The NVU responds to cerebral ischemia with a coordinated inflammatory response through the release of inflammatory mediators. Activation of this inflammatory response allows activation, proliferation, and infiltration of circulating inflammatory cells, such as macrophages, neutrophils, and lymphocytes (Figure 2). Leucocytes will aggravate ischemic lesions and cerebral edema. Many studies already observed the beneficial effects of leucocytes inhibition on cerebral ischemia models. In this section, we study the peripheral inflammation that is a known target of roscovitine in various pathologies.

### 4.1. Macrophages

#### 4.1.1. Pathological Processes

Infiltrating macrophages are highly plastic cells whose phenotype is influenced by their environment and phagocytes [157,158]. In a tMCAO mice model, blood-derived macrophages were recruited in injured tissue from day 3 to 7 after stroke [159], early after ischemia (1 day) microglia predominates [160]. As activated microglia, macrophages infiltrating the ischemic brain are polarized according to their environment. M1 macrophage mainly releases cytotoxic substances, induces inflammation, and leads to cell death, and M2 polarized macrophage releases anti-inflammatory cytokines and promotes tissue remodeling [161]. At the acute phase of brain ischemia, macrophage seems to have biphasic functions. Depending on studies and models of ischemic stroke, macrophage depletion results in a decrease in infarct volume, increase in infarct volume, or no effect at all [162]. These opposites effects may be related to differences between experiments regarding the severity of ischemia, the timing of macrophage polarization switching, or the methods used to inhibit monocytes infiltration into ischemic brain.

#### 4.1.2. Roscovitine and Macrophages

In various models, (*R*)-roscovitine treatment decreased macrophage proliferation and cytokines production, favored M2 polarization in vitro, and decreased infiltration in vivo. However, no studies were conducted on the effect of (*S*)- or (*R*)-roscovitine’s on in vitro or in vivo models of ischemic stroke.

In RAW264.7 cells culture, a monocyte/macrophage-like cell lineage, activated by LPS, (*R*)-roscovitine abolished the production of NO and the expression of iNOS mRNA and protein [163]. (*R*)-roscovitine also decreased phosphorylation of IKKβ, IκB, and p65 but increased phosphorylation of ERK, p38, and JNK. In addition, (*R*)-roscovitine dose-dependently inhibited expression of COX-2, IL-1β, and IL-6 but not TNFα. In isolated peritoneal macrophages, (*R*)-roscovitine inhibited NO production, iNOS, and COX-2 upregulation, and NFκB activation induced by LPS stimulation [163]. In mice bone-marrow-derived macrophage (BMDM) culture stimulated with LPS, (*R*)-roscovitine treatment decreased Il-1β, Il-6, and iNOS mRNA and protein levels [164]. In murine RAW264.7 macrophages culture stimulated by LPS, (*R*)-roscovitine inhibited cell proliferation and diminished nitric oxide production and IL1b, IL6, TNFα, and iNOS expression [163,164,165]. (*R*)-roscovitine suppressed TLR4 macrophage activation and further decreased downstream inflammatory signaling (MyD88, IRF3, p38, JNK, and ERK) [166]. Roscovitine could also have an impact on the polarization of M2 macrophages. In mice BMDMs, treatment with (*R*)-roscovitine before LPS stimulation decreased anti-inflammatory cytokine IL10 mRNA production after 6 h [167].

In vivo, in status epilepticus pilocarpine model in rat, (*R*)-roscovitine treatment decreased monocyte infiltration in the frontoparietal cortex [104].

#### 4.1.3. Specific CDKs Inhibition in Macrophages

Among CDKs inhibited by roscovitine, CDK1, CDK2, CDK5, and CDK7 were studied in macrophages.

##### CDK1 and Macrophages

In vivo, in rat macrophages 3 days after tMCAo, RNA-sequencing data analysis showed an upregulation of genes coding for proliferating markers such as Ki67 and CDK1 as well as other cell cycle proteins [168]. In RAW264.7 cells culture activated by LPS, Du et al. [163] showed that specific inhibition of CDK1 decreased NO production.

##### CDK5 and Macrophages

Mice bone-marrow-derived macrophage (BMDMs) and J774A.1 cells stimulated with LPS induced p35 cleavage in p25 and an increase in CDK5 activity 2h after stimulation [167]. Du et al. [163] tested the effect of a specific inhibitor of CDK2 (CDK2 inhibitor II (compound 3) from Calbiochem) and CDK2/5 (N4-(6-aminopyrimidin-4-yl)-sulfanilamide) on NO production by LPS activated RWA264.7 cells. They found that inhibitors of CDK2 and CDK5, but not CDK2 inhibitor only, decreased NO production, suggesting that the effect of CDK2 and 5 inhibitor was due to CDK5 inhibition. In LPS-stimulated mice BMDMs, specific CDK5 deletion, contrary to (*R*)-roscovitine treatment, did not reduce IL-1β, IL-6, and iNOS mRNA or their protein expression [164]. These results suggested that the effects of (*R*)-roscovitine on pro-inflammatory mediators expression were not mediated by CDK5 inhibition, but probably by the inhibition of other roscovitine targets. However, CDK5 was involved in macrophage anti-inflammatory processes in several models. BMDMs and peritoneal macrophage cultures from p35-KO mice stimulated by LPS generated more anti-inflammatory IL-10 mRNA 4h after stimulation and more IL-10 protein 2 days after stimulation than WT [167]. Similarly, J774A.1 cells and BMDMs transfected with CDK5 siRNA showed an increased IL-10 mRNA and protein expression compared to controls in response to LPS [167]. In BMDMs from CDK5-KD mice, stimulation for 24 h with the M2 stimuli IL-4, IL-10, and IL-13 had no impact on M2-like markers (Cd163, Cd206, Ym1, and Il-10) or phagocytosis receptors (Cd36, Anxa1) [164], suggesting that CDK5 was necessary for M2 polarization.

##### CDK7 and Macrophages

In RAW264.7 cells culture activated by LPS, specific inhibition of CDK7 with 5,6-dichlorobenzimidazole 1-β-d-ribofuranoside decreased NO production [163].

#### 4.1.4. Conclusions

In various models, (*R*)-roscovitine decreases macrophage proliferation and cytokines production, favors M2 polarization in vitro, and decreases infiltration in vivo. CDK1 is associated with macrophage proliferation in vivo after brain ischemia. In other inflammatory models in vitro, CDK1, -5, and -7 are associated with NO production, and CDK5 is involved in M2 polarization and IL10 secretion.

### 4.2. Neutrophils

#### 4.2.1. Pathological Processes

Neutrophils intravascular adhesion is rapid after ischemic stroke, but parenchymal infiltration is usually observed later. In most studies, neutrophils are the first cells to invade the brain parenchyma, 48 to 72 h post-stroke in most models, and their population declined rapidly thereafter [169,170]. Neutrophils contribute significantly to post-ischemic inflammation and tissue lesions [171]. Their deleterious role includes limiting tissue perfusion by intravascular occlusion [172], releasing MMPs that destabilize the BBB [173], generating ROS and reactive nitrogen species (RNS) [174]. Interestingly, the principal neutrophils action site after brain ischemia seems to be the BBB, where neutrophils are found in greater numbers than in brain parenchyma [31,175]. The transition of inflammation toward its resolution involves elimination of neutrophils in the brain parenchyma [176].

#### 4.2.2. Roscovitine and Neutrophils

To our knowledge, no studies have been conducted on (*S*)- or (*R*)-roscovitine’s effect on in vitro or in vivo model of ischemic stroke. However, both were studied in non-ischemic conditions. (*R*)-roscovitine’s effects on CDKs gene expression and protein level in human neutrophils was studied by Leitch et al. [177]. (*R*)-roscovitine treatment decreased CDK2, CDK7, and CDK9 gene expression in both unstimulated and LPS-stimulated isolated human neutrophils, but had no effect on other CDK1, 3, 4, 5, and 8 genes expression. Among cyclin D1, cyclin H, and cyclin T1, binding partners, respectively, of CDK2, CDK7, and CDK9, only cyclin H gene expression decreased after (*R*)-roscovitine treatment in both unstimulated and stimulated neutrophils [177]. CDK5 protein level increased in the nuclear fraction of neutrophils after LPS stimulation was inhibited by (*R*)-roscovitine. No effect of (*R*)-roscovitine in CDK7 or CDK9 proteins level was observed after LPS stimulation; however, (*R*)-roscovitine inhibited transcriptional machinery induced by CDK7 and CDK9 [177].

Both (*R*)- and (*S*)-roscovitine induced neutrophil apoptosis in vitro. In isolated human neutrophils, (*R*)- and (*S*)-roscovitine both induced neutrophil apoptosis at similar degree 6 h after treatment (75.9 ± 3.5% for (*R*)-roscovitine and 75.6 ± 3.3% for (*S*)-roscovitine) [178]. Interestingly, at 20h, the pro-apoptotic effect of (*R*)-roscovitine was dominant over survival factor GM-CSF (granulocyte-macrophage colony-stimulating factor). (*R*)-roscovitine increased caspase-3 cleavage in neutrophils and then increased caspase activation. CDK1 and CDK2 are both present in human neutrophils, and their protein levels remained stable during apoptosis with and without (*R*)-roscovitine. However, CDK1 activity decreased rapidly during apoptosis [178]. Other authors showed that (*R*)-roscovitine was able to override survival mediators TNFα and LPS to induce apoptosis in human neutrophils [179,180]. Further studies in mouse neutrophils and neutrophil progenitor cells culture showed that (*R*)-roscovitine promoted apoptosis by reducing concentrations of the anti-apoptotic proteins like Mcl-1, Bim, Puma, or Noxa [178,181]. (*R*)-roscovitine also blocked neutrophils degranulation in vitro. In humans, for permeabilized neutrophils stimulated by GTP, (*R*)-roscovitine blocked neutrophil degranulation and lactoferrin secretion [182]. In mice, for isolated neutrophils stimulated with GTP, (*R*)-roscovitine also decreased the surface expression of granule secretion marker CD63 and CD66b [182]. Furthermore, in humans, fir isolated neutrophils, (*R*)-roscovitine treatment before GTP stimulation reduced phosphorylation of vimentin Ser56 and secretion of β-hexosaminidase, lactoferrin, and MMP-9 by cells compared to WT [183].

In vivo, an in-mouse model of carrageenan-induced pleural inflammation, (*R*)-roscovitine intraperitoneal injection before carrageenan intrapleural injection inhibited the total pleural inflammatory cell number by more than 50%, increased total apoptotic cells number and reduced number of neutrophils compared with vehicle [178]. In the bleomycin-induced lung injury model in mice, (*R*)-roscovitine administration after inflammation induction by bleomycin, reduced the total number of neutrophils in the bronchoalveolar fluid after 3 days [178] and 7 days [177]. (*R*)-roscovitine also decreased mRNA expression of inflammatory cytokines (IL1, GMCSF, CINC-1) and protein level of NFkB and COX-2 by neutrophils in a endotoxin-induced uveitis mouse model [184].

#### 4.2.3. Specific CDKs Inhibition in Neutrophils

No studies were conducted on CDKs functions in neutrophils in ischemic stroke, nevertheless, it was widely studied in non-ischemic conditions [176]. In-vitro, using genechip technology, Leitch et al. [177] showed that human isolated neutrophils expressed gene of CDK1, -2, -3, -4, -5, -6, -7, -8, -9, and -10, at least, but CDK2, CDK7, and CDK9 were the most strongly expressed. The same authors showed by Western blot that CDK5, -7, and -9 were highly expressed, but CDK2 protein levels were low. CDK5, CDK7, and CDK9 were detected equally in both cytoplasmic and nuclear fractions [177]. Leitch et al. [177] did not study CDK1 protein expression on neutrophils; however, other authors [178,185] showed that both proteins CDK1, -2, -5, -7, and -9 were expressed in human isolated neutrophils.

##### CDK1 and Neutrophils

In human isolated neutrophils, CDK1 protein level remained stable during apoptosis induced by the activating Fas antibody CH11 [178]. However, CDK1 activity decreased rapidly after apoptosis induction, suggesting that shutdown of CDK1 activity is necessary for neutrophil apoptosis.

##### CDK2 and Neutrophils

Expression of CDK2 was downregulated during neutrophil differentiation in human [186]. CDK2 protein level was clearly expressed in the most immature population’s myeloblasts and promyelocytes and strongly decreased in more mature myelocytes and metamyelocytes stage, and no CDK2 expression was detected in mature polynuclear neutrophils from peripheral blood.

##### CDK5 and Neutrophils

In human isolated neutrophils, Lee et al. [183] showed that CDK5 regulated secretion of pro-inflammatory molecules. After stimulation of neutrophils with GTP, they observed an increased phosphorylation of vimentin at Ser56 and an increased colocalization with CDK5, in association with increased secretion of β-hexosaminidase, lactoferrin, and MMP-9. The phosphorylation of vimentin at Ser56 resulted in vimentin depolymerization and disassembly of neutrophil cytoskeleton, thereby allowing the release of neutrophil vesicular contents. CDK5 siRNA transfection before GTP stimulation reduced phosphorylation of vimentin Ser56 and secretion of β-hexosaminidase, lactoferrin, and MMP-9 from cells compared to WT [183]. Rosales et al. [182] found in human isolated neutrophils that formation of a CDK5-p35 complex, and CDK5 activity, were more important in the granule than in the membrane fraction, suggesting that CDK5-p35 activity in neutrophils was associated with the secretion of granule contents. Coherently, GTP stimulation provoked an increase of CDK5 activity in the granule and an increase in lactoferrin secretion. Furthermore, in neutrophils isolated from p35-KD mice and stimulated with GTP, authors [182] observed a decrease in surface expression of granule secretion marker CD63 and CD66b compared to WT.

In a mouse model of induced experimental autoimmune encephalomyelitis, CDK5-null immune chimeric mice showed a reduced infiltration of neutrophils in CNS compared to WT [187].

##### CDK9 and Neutrophils

CDK1/2/5 inhibitor NU6102 had no effect on human neutrophil apoptosis, but CDK1/2/5/7/9 inhibitor (*R*)-roscovitine induced apoptosis and CDK1/2/5/9 inhibitor LGR 1406 increased apoptosis, suggesting that CDK9 inhibition is responsible for apoptosis induction of both (*R*)- and (*S*)-roscovitine [185]. Consistently, authors also observed that CDK9 activity and cyclin T1 expression decreased as neutrophils aged in culture and entered spontaneously in apoptosis [185]. In vivo in zebrafish, targeting CDK9 by CDK9-KD or CDK9-KO both increased neutrophils apoptosis [188].

#### 4.2.4. Conclusions

(*R*)-roscovitine in vivo and both (*S*)- and (*R*)-in vitro are able to induce neutrophils apoptosis. (*R*)-roscovitine also decreases degranulation in vitro. CDK1, -2, -5, -7 and -9 are present in human neutrophils at different stages. However, their role after ischemic stroke is not studied yet. In neutrophils, CDK2 is associated with neutrophil differentiation, while CDK5 with degranulation and CDK9 with apoptosis.

### 4.3. Eosinophils

#### 4.3.1. Pathological Processes

The relationship between eosinophils and acute ischemic stroke (AIS) is unclear and was rarely studied. Eosinophils might be associated with a better outcome after ischemic stroke. A recent study revealed a negative correlation between eosinophil number and NIHSS score (National Institute Of Health Stroke Score) in patients with AIS [189]. Another pilot study including 973 patients with an ischemic stroke showed that patients with higher eosinophil counts recovered faster and experienced less functional impairment in limbs than patients with lower eosinophil counts [190]. A retrospective study on 201 acute ischemic stroke patients who were treated with rtPA (recombinant tissue plasminogen activator) within 4.5 h of symptom onset suggested that absolute eosinophils count (AEC) on admission could be considered an as independent predictive marker of hemorrhagic transformation after treatment with rtPA. In this study, they showed that higher values of AEC (≥ 0.11 × 109/l) were independently associated with a 78% reduction in the odds of developing hemorrhagic transformation [191]. Eosinopenia was also associated with large-infarct-volume patients with first acute ischemic stroke [192]. This study showed a strong correlation between the number of eosinophils and the severity of ischemic stroke; however, no causal relationship was yet demonstrated.

#### 4.3.2. Roscovitine and Eosinophils

No studies on (*S*)- or (*R*)-roscovitine’s effect on eosinophils on ischemic stroke models have been described. A few studies, however, tested (*R*)-roscovitine’s effect on eosinophils in non-ischemic conditions [176]. In vitro, (*R*)-roscovitine induced human blood eosinophil apoptosis in a time- and concentration-dependent manner, associated with a suppression of Mcl-1L expression and an enhanced phagocytic clearance of eosinophils by macrophages [180]. (*R*)-roscovitine treatment promoted human eosinophil apoptosis by activating caspases, inducing loss of mitochondrial membrane potential and downregulation of key survival protein Mcl-1 [193]. (*R*)-roscovitine also induced apoptosis in mice eosinophils cultures derived from peripheral blood, spleen, and bone marrow [180]. Furthermore, as for neutrophils, (*R*)-roscovitine was shown to decrease degranulation in freshly isolated human eosinophils [194]. Only one study was published about the effect of (*R*)-roscovitine on eosinophils in vivo and showed that (*R*)-roscovitine induced a non-significant decrease of eosinophils cell count in an ovalbumin mouse model of human asthma [180].

#### 4.3.3. Specific CDKs Inhibition in Eosinophils

It was shown that human blood eosinophils expressed CDK1, -2, -5, -7, and -9 at both mRNA and protein levels [180]. In isolated human eosinophils, CDK5 and p35 were expressed, and CDK5 kinase activity was observed after eosinophil stimulation by calcium ionophore, platelet-activating factor, or eotaxin/CCL11 [194]. siRNA knock-down of CDK5 expression decreased eosinophils degranulation [194], suggesting that CDK5 regulated eosinophils degranulation, maybe through the same pathway as neutrophils.

#### 4.3.4. Conclusions

Some studies showed a positive correlation between eosinophils number and ischemic stroke outcomes. (*R*)-roscovitine is associated with eosinophils apoptosis and inhibition of degranulation in different models but not in ischemia. CDK1, -2, -5, -7 and -9 are found in human eosinophils. CDK5 regulates eosinophil degranulation in vitro.

### 4.4. T Lymphocytes

#### 4.4.1. Pathological Processes

After focal ischemia T-lymphocytes infiltration takes place on day 1, increases between days 3 to 7 [195,196], and is still observed on day 14 [197,198,199]. There are several T cell types, including T helper cells (CD4+), killer T cells (CD8+), regulatory T cells (FoxP3), or γδT cells involved in different pathological processes during stroke. CD4+ T cells differentiate in Th1, Th2, Th17, or iTreg (induced regulatory T cells), depending on external stimuli (Figure 3). Th1 and Th17 cells may aggravate brain injury by secreting pro-inflammatory cytokines, while Th2 and Treg cells may have neuroprotective effects on the injured brain by secreting anti-inflammatory cytokines [161]. T cells mainly play a detrimental role in the acute phase of ischemic stroke. On various stroke models, transgenic animals lacking T cells, or animals with antibody-mediated depletion of CD4+, CD8+, or γδT cells presented ischemic damage reduction compared to corresponding WT [199,200,201,202]. Conversely, regulatory T cells appear in ischemic tissue after the acute phase and could have a protective effect by decreasing post-ischemic inflammation. They limit ischemic injury extension and counteract excessive proinflammatory cytokines expression, notably through IL-10 secretion, infiltration modulation, and activation of lymphocytes and microglia after brain ischemia [203,204]. Tregs isolated from wild type mice transferred into stroke mice 2 h after 60 min of tMCAO model decreased BBB disruption 24 h after ischemia and decreased both infarct volume and brain edema 3 days after ischemia [205]. However, the intravascular presence of Treg during the post-tMCAO reperfusion phase may contribute to ischemic damage by causing vascular dysfunction and thrombus [206].

#### 4.4.2. Roscovitine

Many studies have been conducted to study the effect of (*R*)-roscovitine on T lymphocytes. However, no study on roscovitine after brain ischemia was identified. (*R*)-roscovitine reduced activation and cytokine production, and induced apoptosis of CD4+ T cells isolated from mice and activated in vitro by anti-CD3 and anti-CD28 antibodies [207]. In mouse T cells, (*R*)-roscovitine treatment decreased IL-2 secretion and mRNA expression following stimulation with anti-CD3/CD28 [208]. In mouse-purified T cells activated with anti-CD3/CD28, (*R*)-roscovitine dose-dependently inhibited cells proliferation, expression of surface activation marker CD69, and secretion of IL-2, IFNγ, and TNFα [209]. In these cells, (*R*)-roscovitine prevented CDK2 phosphorylation and downregulation of p27, p-Rb, and cyclin A, suggesting that cell cycle was blocked in G1/S. In mouse, for purified T cells treated with TNFα, (*R*)-roscovitine regulated TNFα-mediated NFκB activation, suggesting that its effect on cytokine secretion was due to NFκB blocking [209]. In mouse, for total splenocyte and isolated T cell stimulated with CD3/CD28, PMA/ionomycin, or allogeneic dendritic cells, (*R*)-roscovitine also strongly decreased T cell proliferation [187]. In rat lymph node cells activated by Concanavalin A in vitro, (*R*)-roscovitine decreased the proliferation of lymph node cells and CD8+ T cells activation [210]. Another study showed that (*R*)-roscovitine was able to inhibit lymphocyte migration toward the chemokine CCL19 [187].

In vivo, transfer of T cells treated in vitro by (*R*)-roscovitine in an ovalbumin-induced uveitis mice model decreased rolling and adherent cells in the vasculature of the iris compared to WT T cell injection [207]. In mice treated by (*R*)-roscovitine in vivo, splenocytes were isolated from mice and activated ex vivo by Concanavalin A or anti-CD3/CD28. In both stimulation methods, a reduced T cell proliferation and IFNγ and IL-10 production was observed in the treated group compared to vehicle [211]. In a graft-versus-host disease model in mice, irradiated mice received bone marrow cells and splenocytes transplantation and then were treated or not with (*R*)-roscovitine. (*R*)-roscovitine treatment blocked T cells expansion 7 days and 3 weeks after transplantation [209].

#### 4.4.3. Specific CDKs Inhibition in T Cells

No studies have been conducted yet on T lymphocytes CDKs in ischemic stroke. Nevertheless, we can learn from a large panel of studies about CDKs’ functions in T lymphocytes.

##### CDK1 and T Lymphocytes

Fas-induced apoptosis is a major mechanism of T cell cytotoxicity [212,213]. Fas and Fas-Ligand are overexpressed in ischemic stroke patients and were associated with apoptosis [214]. Delivery of a Fas-blocking peptide attenuated Fas-mediated apoptosis in brain ischemia [215]. In T cells activated in vitro by anti-CD3, cyclin B1 inhibition, partner of CDK1, by cyclin B1 antisense oligonucleotides inhibited Fas-Ligand expression [216]. Using DN-CDK1, authors showed that CDK1/cyclin B1 activity induced Fas-Ligand transcription through the regulation of NFκB activation, suggesting that CDK1 was involved in Fas-Ligand cytotoxic pathway.

##### CDK2 and T Lymphocytes

CDK2 was associated with T cell activation and cytokine secretion in vitro. In isolated CD4+ T cells from mice, activation by anti-CD3 and anti-CD28 antibodies induced CDK2 expression [207]. CD4+ T cells from CDK2-deficient mice stimulated with anti-CD3 did not show proliferation level variation compared to WT T cells; however, they showed a decreased secretion of IL2 and IFNγ [217].

##### CDK5 and T Lymphocytes

Several publications associated CDK5 with T cell activation, proliferation, migration, and cytokine secretion, suggesting that CDK5 is a key regulator of T cell physiology. In isolated mouse T cells, CDK5 and p35 were found in very low abundance [187]. However, T cell stimulation with CD3/CD28 increased both CDK5 and p35 protein and mRNA expression, associated with a strong increase in CDK5 kinase activity. Interestingly, stimulation of T cells with splenocytes protein lysate from EAE mice also strongly increased CDK5 activity. T cells from CDK5-null immune chimeric mice (CDK5^−/−^ T cells) and p35^−/−^ T cells stimulated with CD3/CD28 showed a great reduction of proliferation compared to WT, suggesting that CDK5 regulated T cell proliferation [187]. The authors further showed that CDK5^−/−^ T cells present deficient IL-2 secretion and mRNA expression following stimulation with anti-CD3/CD28 [208]. They observed that CDK5 modulated gene expression by impairing the repression of gene transcription by histone deacetylase 1 (HDAC1).

In vivo, in mouse experimental autoimmune encephalomyelitis (EAE) model, CDK5-null immune chimeric mice showed a reduced infiltration of T cells in CNS compared to WT [187]. CDK5^−/−^ lymphocytes showed a lower coronin1a phosphorylation level than WT. Immunoprecipitation confirmed that CDK5 interacted directly with coronin1a and phosphorylated it. Disruption of CDK5 activity and therefore coronin1a phosphorylation in T cell CDK5^−/−^ also impaired actin polarization and migration of lymphocytes to CCL19 signals, suggesting that CDK5 may control CDK5 activation and migration [187].

##### CDK9 and T Lymphocytes

In T cells isolated from healthy donors, CDK9 and Cyclin T1 protein and mRNA expression level was lower in naïve T cells and increased in memory T cells, effector T cells, and particularly activated T cells, suggesting that CDK9 was involved in T cells differentiation [218]. Naïve T cells stimulated to differentiate by different cocktails of growth factors showed an increase in CDK9 mRNA and cyclin T1 mRNA one day after stimulation, supporting the idea that CDK9/cyclin T1 was involved in T cells differentiation [218].

#### 4.4.4. Th17/Treg Lymphocytes Imbalance

The differentiation of Th17 cells from naïve T cells requires TGFβ and IL-6, which induced RORγt expression (Figure 3). Th17 cell is characterized by pro-inflammatory cytokines secretion, such as IL17A. In contrast, Foxp3, the master regulator of Tregs, is induced by TGFβ and IL-2, and Treg secretes anti-inflammatory cytokines such as TGFβ and IL-10. Th17 and iTregs reciprocally inhibit their differentiation. In patients at 1, 5, and 10 days after ischemic stroke, a significant reduction of peripheral Treg cell frequency and TGFβ and Foxp3 expression levels were observed, while the proportions of Th17 were increased dramatically, with increased levels of IL-17A and RORγt expression [219]. Th17 cells and their signature cytokine IL-17 are associated with cognitive impairment in a wide variety of neurological diseases, including ischemic brain injury, multiple sclerosis (MS), and Alzheimer’s disease [220]. In post-mortem brain tissue of patients who died within 24 h of stroke, IL-17A-positive T cells were detected in the infarcted area, suggesting their involvement in ischemic cascade injury after stroke [201]. Gelderbloom et al. [201] suggested that selective targeting of IL-17A signaling might provide a new therapeutic option for the treatment of stroke. They showed that IL-17A-blocking antibody injected 3 h after stroke induction decreased infarct size and improved neurologic outcome in tMCAo mouse model.

##### Roscovitine in Th17/Treg Lymphocytes Imbalance

Interestingly, Yoshida et al. [221] screened 285 chemical inhibitors of Th17 differentiation; among them, they found roscovitine. (*R*)-roscovitine suppressed Th17 differentiation and enhanced iTreg development in vitro [221] (Figure 3). In vivo, administration of (*R*)-roscovitine to mice on EAE model suppressed T naive cell differentiation in pro-inflammatory Th17 cells and enhanced differentiation in anti-inflammatory Treg cells by reinforcing FOXP3 expression [221]. Yoshida et al. [221] found on Th17 cell culture that (*R*)-roscovitine enhanced STAT5 phosphorylation, a strong inhibitor of Th17 and promoter of iTreg differentiation, and restored IL2 production, the main activator of STAT5 in T cell, suggesting that (*R*)-roscovitine suppressed Th17 by inhibiting TGFβ-mediated IL-2 suppression. Another essential kinase regulating the differentiation of Th17 and Tregs is DYRK1A [222], an (*S*)- and (*R*)-roscovitine target [223]. Inhibition of DYRK1A enhances Treg differentiation, impairs Th17 differentiation, and attenuates inflammation [222]. Interestingly, in CD4+ T cell isolated from mice, (*R*)-roscovitine reduced the secretion of IL-17 and IFNγ induced by anti-CD3/CD28 [207].

##### CDKs in Th17/Treg Lymphocytes Imbalance

CDK2 seems to play a role in the differentiation of naïve T cells to Th17 or Treg. CDK2–cyclin E can phosphorylate Foxp3 in T cells and then inhibit Treg differentiation [217]. In mice transplanted with cardiac allografts, CDK2-deficient mice exhibited an increased infiltration of Foxp3+ Tregs than WT mice in surviving grafts. In vitro, isolated CDK2-/- Treg showed a stronger capacity to suppress proliferation of naïve T cells than WT Treg. Together, these observations suggest that CDK2 promotes conventional T cell differentiation and restricts Treg function [217].

#### 4.4.5. Conclusions

(*R*)-roscovitine decreases T lymphocytes activation, proliferation, and migration in several in vitro and in vivo models, but it was not shown in cerebral ischemia models. (*R*)-roscovitine also promotes Treg lymphocytes polarization in vitro and in vivo but was not studied in models of focal ischemia. In T lymphocytes, CDK1 is associated with Fas-mediated apoptosis; CDK2 with lymphocytes activation and lymphocytes Th17 differentiation; CDK5 with lymphocytes activation, proliferation, migration, and cytokine secretion; and finally CDK9 with lymphocytes differentiation.

### 4.5. B Lymphocytes

#### 4.5.1. Pathological Processes

The role of B cells in ischemic stroke is not still clearly understood, but some authors reported beneficial effects of B cells control after brain ischemia. Ren et al. [224] in an MCAo mice model observed that B-cells-deficient mice have larger infarct volumes, higher mortality, more severe functional deficits, and increased numbers of activated T cells, macrophages, microglial cells, and neutrophils in the affected brain hemisphere than WT mice at 48 h. This beneficial effect was due to the secretion of neuroprotective IL-10 by B-cells [224]. Treatment with therapeutic IL-10-secreting B cells, injected at 24 h after MCAO, resulted in reduced infarct volumes and improved neurological deficits [225]. Recently, Ortega et al. [226] also showed that IV B cells transfer to mice reduced infarct volume at 3 and 7 days and that this effect was mediated by IL-10 secretion. In contrast, other authors showed that B cells did not have a major pathophysiologic role in acute ischemic stroke in mice [200,227,228]. These differences might be attributed to ischemia severity or time of observation.

#### 4.5.2. Roscovitine

There are very few studies on roscovitine effects on B cells, and none in ischemic stroke. In vitro, several studies showed that (*R*)-roscovitine triggers apoptosis in B-cell chronic lymphocytic leukemia cells [229,230]. In isolated splenocytes from (*R*)-roscovitine-treated mice, a reduced B cell proliferation and IgG2a release was observed in the (*R*)-roscovitine-treated group compared to vehicle after activation by anti-CD3/CD28 [211].

#### 4.5.3. Specific CDKs Inhibition in B-Lymphocytes

No studies were conducted on CDKs function in B cells during ischemia. However, CDKs functions were studied in non-ischemic conditions. In three B-cell lines (Ramos, Reh-6, and IA), CDK1 protein and mRNA levels varied according to the cell cycle phase [231,232]. In primary cultures of murine B cells, CDK2, and cyclin A were not detectable in B cells G0 and G1 but were expressed during the S phase [231]. Specific inhibition of CDK2, CDK5, and CDK7 was tested in B-cells lymphoma cell lines. CDK2 inhibition by CVT-313 treatment or CDK2 siRNA induced apoptosis [233], CDK5 inhibition by CDK5-specific shRNAs reduced proliferation and increased apoptosis [234], and CDK7 inhibitor QS1189 induced apoptosis and cell cycle arrest [235].

CDK9 was also strongly studied in isolated human B cells from healthy donors. CDK9 and cyclin T1 protein and mRNA levels were higher, respectively, in memory B cells versus naïve B cells, and in activated B cells versus non-activated ones [236]. In human naïve B cells from peripheral blood stimulated in vitro by different cocktails of growth factors to induced differentiation, expression level of the CDK9/Cyclin T1 complex did not increase. However, CDK9 interacted with E12 and E47 and was co-localized in the germinal center of the lymph node. E12 and E47 are members of the helix–loop–helix family, which drive differentiation in lymphoid tissue. Thus, these results suggested an active role for CDK9/Cyclin T1 complex during B cell differentiation [236].

#### 4.5.4. Conclusions

(*R*)-roscovitine triggers lymphocytes B apoptosis in vitro and decreases B lymphocytes proliferation in vivo in inflammatory models, but no study was conducted on ischemic stroke. There is a lack of knowledge of CDKs’ functions in B lymphocytes after ischemia. However, in vitro, CDK2 and CDK5 are associated with proliferation and apoptosis of B lymphocytes, while CDK7 is associated with apoptosis and cell cycle arrest and CDK9 with B lymphocytes differentiation.

## 5. Conclusions

R/S -roscovitine showed a beneficial effect in several models of ischemic stroke. (*S*)-roscovitine decreases brain edema and infarct volume, and this effect has already been associated with a decrease in neuronal death, BBB protection, endothelial protection, decrease in microglial proliferation, and astrocyte reactivity. Several studies indicate that (*R*)-roscovitine acts on other NVU cells, as well as leucocytes in non-ischemic conditions. In several models, specific inhibition of roscovitine targets, CDK1, -2, -5, -7, and -9, showed that they are involved in a large panel of processes on the NVU and leucocytes. Our review supports the investigation of (*S*)-roscovitine and specific CDKs inhibition as potential therapeutic agents for the treatment of ischemic stroke.

## Figures and Tables

**Figure 1 cells-10-00104-f001:**
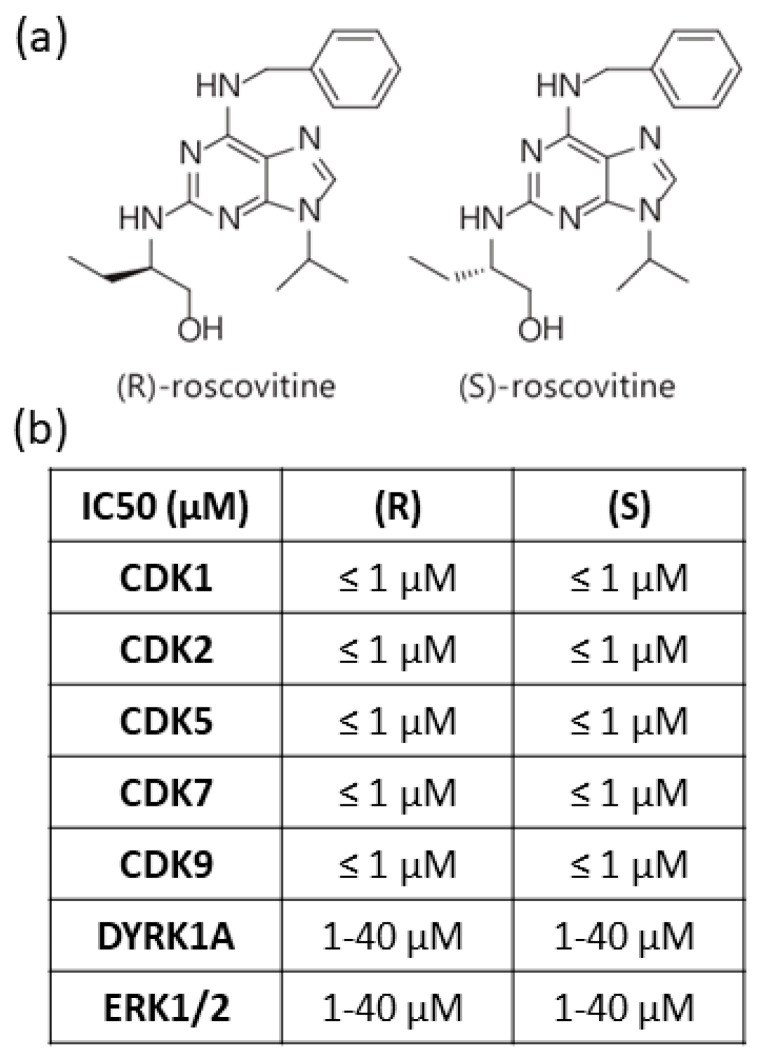
(*R*)-roscovitine and (*S*)-roscovitine: structure and activity. (**a**) Structure of (*R*)- and (*S*)- stereoisomer of roscovitine. (**b**) IC50 values (in µM) described for these compounds.

**Figure 2 cells-10-00104-f002:**
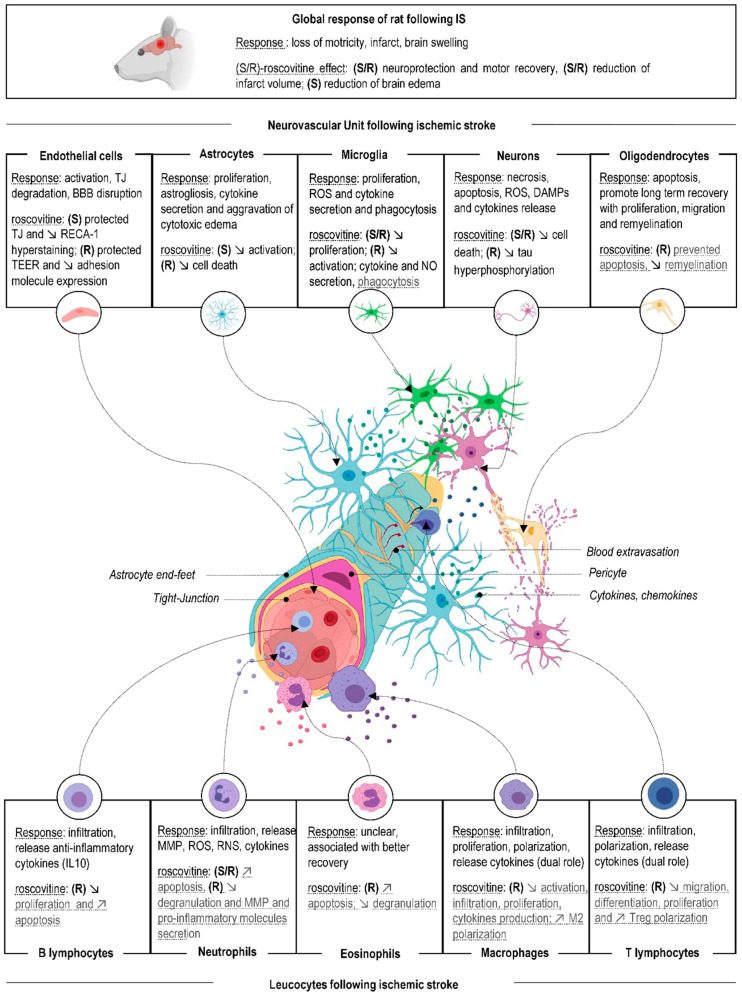
(*R*) and (*S*)-roscovitine effect on Neurovascular Unit (NVU) and immune cells following ischemic stroke. A box was assigned to each cell from the NVU, in which two topics are described: the cell response after ischemic stroke, and the cellular effect of (*R*) and (*S*)-roscovitine. The box content is a summary of the data described for each cell type. For more detailed information, please refer to the full text of the manuscript. The (*R*)-roscovitine is represented by (*R*) and the (*S*)-roscovitine by (*S*). Black text means that the effect of roscovitine was observed in ischemic stroke model in vitro or in vivo. Grey text with dashed highlights means that the (*R*) or (*S*)-roscovitine effect was observed in other non-ischemic models, in vitro or in vivo. Created with BioRender.com.

**Figure 3 cells-10-00104-f003:**
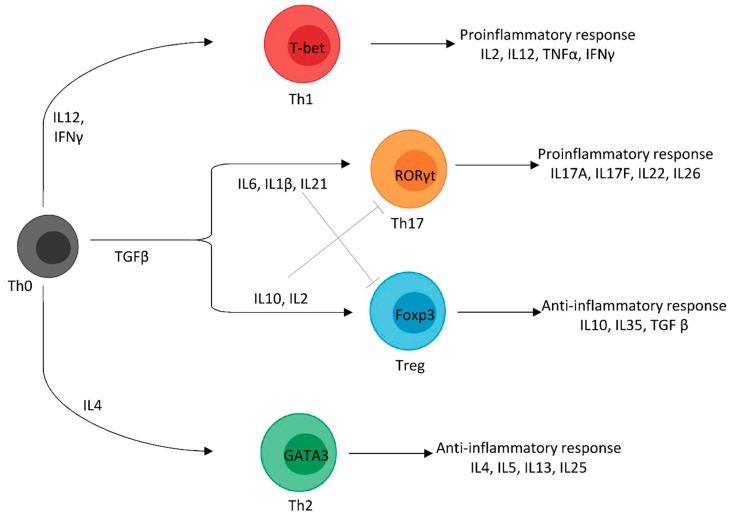
Lymphocyte Th17/Treg balance. CD4+ T cells differentiate in Th1, Th2, Th17, or iTreg (induced regulatory T cells) depending on external stimuli. Th1 cells may aggravate brain injury by secreting pro-inflammatory cytokines (through T-bet), while Th2 cells may have neuroprotective effects on the injured brain by secreting anti-inflammatory cytokines (through GATA-3). The differentiation of Th17 cells from naïve T cells requires TGF-β and IL-6, which induce RORγt expression. Th17 cell is characterized by pro-inflammatory cytokines secretion, such as IL17A. In contrast, Foxp3, the master regulator of Tregs is induced by TGF-β and IL-2, and Treg secretes anti-inflammatory cytokines such as TGF-β and IL-10. Th17 and iTregs reciprocally inhibit their differentiation. Created with BioRender.com.

## Data Availability

Data sharing not applicable.

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
