# Peer review of "Cellular and Molecular Mechanisms of R/S-Roscovitine and CDKs Related Inhibition under Both Focal and Global Cerebral Ischemia: A Focus on Neurovascular Unit and Immune Cells"

_cells, 2021, doi:10.3390/cells10010104_

Round 1

Reviewer 1 Report

The manuscript titled as “Effect of R/S -roscovitine and CDKs specific inhibition on neurovascular unit and leucocytes after focal cerebral ischemia.” by Roy L.L. et., al. to the Cells has great potentiality and might help in the search of a potential therapeutic agent for cerebral ischemia. However, there are still some major concerns and modification need to be done to increase the impact of the manuscript.

  1. The first concern of this article is the title of this article. A large part of the manuscript discussed about the effect of roscovitine on non-ischemic disease. However, the title reflects that this manuscript is only centered to focal ischemic injury. The title needs to be changed and make it more appropriate according to the body text of the manuscript. Even the term “focal Ischemia” doesn’t include all the cerebral ischemia discussed in this manuscript.
  2. In the introduction, authors discussed more about the common topics of ischemic injury and didn’t introduce roscovitine and CDK at all. Authors should introduce roscovitine and CDK in the introduction part. In the CDK part the authors discussed only CDK5 elaborately- why? Why other CDKs did not get emphasis?
  3. The authors did not discuss about the NVU and its relationship with cerebral ischemia. They should add a separate part introducing NVU along with its components which is discussed later.
  4. The authors should explain the necessity of the inclusion of the different immune cells (macrophage, neutrophils etc) as the authors mentioned that there are no studies relating the roscovitine and ischemic stroke.
  5. In the specific cellular parts, the authors discussed the pathological process of the cell in ischemic injury and the role of roscovitine is discussed in non-ischemic studies which is very confusing and seems does not necessary in this review. If the authors still want to discuss the role of roscovitine they should discuss the pathological process of the specific cells in those non-ischemic disease.
  6. Overall, the manuscript needs extensive editing and make the discussed information more related to the topic and title intended.
  7. Figure 2 and 3 seems to be too large and two figures are seeming to be un-necessary. The authors can use one cartoon to explain the NVU and the immune cell response in one well organized figure.
  8. Proper extensive language editing in needed.

Author Response

Dear Sir, Dear Madam,

thank you for your comments. We have taken in account all your comments.

Sincerely Yours,

Serge Timsit

Reviewer 2 Report

The manuscript entitled “Effect of R/S-roscovitine and CDKs specific inhibition on neurovascular unit and leucocytes after focal cerebral ischemia” The manuscript entitled “Effect of R/S-roscovitine and CDKs specific inhibition on neurovascular unit and leucocytes after focal cerebral ischemia” reviewed molecular and cellular mechanisms of R- and S-roscovitine and CDKs inhibition on neurovascular unit and leukocytes following ischemic stroke. The draft of this review has been logically written, and the contents have been systematically composed, which can easily deliver the main theme of the present manuscript to readers. However, this reviewer noticed some minor problems which should be addressed for potential publication. The points were summarized below.

1. The present manuscript is dealing with molecular and cellular mechanisms of roscovitine and CDKs inhibition under both focal and global cerebral ischemia. In this regard, this reviewer recommends revising the title of the present manuscript into comprehensive title which can encompass those mechanisms under ischemic stroke.

2. There are some omissions of references. Please cite appropriate references in corresponding sentences as follows:
1) “Introduction” section: line 40-41, line 47-49 and line 70-77
2) “Cyclin Dependent Kinase” section: line 85-88
3) “(R)-Roscovitine and neurons” section: line 316-317
4) “Roscovitine in other non-ischemic models” section: line 401-404
5) “Specific CDKs inhibition in oligodendrocytes” section: line 543-550

3. The section has been incorrectly numbered from the “Effects of roscovitine on Neurovascular Unit” section. This section should be numbered as 5 and the "Effects of roscovitine on leukocytes" section should be numbered as 6.

4. This reviewer regards that the language style of the present manuscript is appropriate; however, this reviewer recommend that the authors should check throughout the manuscript minor points of language such as typos and omissions.

Author Response

Dear Madam, Dear Sir,

thank you for your comments. We have taken in account all your comments. Please see the attachment for the answers.

Sincerely Yours,

Serge Timsit

Round 2

Reviewer 1 Report

Overall, the edits have strengthened the conclusions presented in the manuscript.